# NMR Based Metabolomics Comparison of Different Blood Sampling Techniques in Awake and Anesthetized Rats

**DOI:** 10.3390/molecules24142542

**Published:** 2019-07-12

**Authors:** Hongying Du, Shuang Li, Yingfeng Zhang, Huiling Guo, Liang Wu, Huili Liu, Anne Manyande, Fuqiang Xu, Jie Wang

**Affiliations:** 1College of Food Science and Technology, Huazhong Agricultural University, Wuhan 430070, China; 2Key Laboratory of Magnetic Resonance in Biological Systems, State Key Laboratory of Magnetic Resonance and Atomic and Molecular Physics, Wuhan Institute of Physics and Mathematics, Chinese Academy of Sciences, Wuhan 430071, China; 3University of Chinese Academy of Sciences, Beijing 100049, China; 4School of Human and Social Sciences, University of West London, Middlesex TW89GA, UK

**Keywords:** blood collection, serum, NMR, metabolomics, saphenous vein

## Abstract

The composition of body fluids has become one of the most commonly used methods for diagnosing various diseases or monitoring the drug responses, especially in serum/plasma. It is therefore vital for investigators to find an appropriate way to collect blood samples from laboratory animals. This study compared blood samples collected from different sites using the NMR based metabolomics approach. Blood samples were collected from the saphenous vein (awake state), tail vein (awake and anesthetized states after administration of sevoflurane or pentobarbital) and the inferior thoracic vena cava (ITVC, anesthetized state). These approaches from the saphenous and tail veins have the potential to enable the collection of multiple samples, and the approach from ITVC is the best method for the collection of blood for the terminate state. The compositions of small molecules in the serum were determined using the ^1^H-NMR method, and the data were analyzed with traditional correlation analysis, principle component analysis (PCA) and OPLS-DA methods. The results showed that acute anesthesia significantly influenced the composition of serum in a very short period, such as the significant increase in glucose, and decrease in lactate. This indicates that it is better to obtain blood samples under the awake state. From the perspective of animal welfare and multiple sampling, the current study shows that the saphenous vein and tail vein are the best locations to collect multiple blood samples for a reduced risk of injury in the awake state. Furthermore, it is also suitable for investigating pharmacokinetics and the effects of drug intervention on animals.

## 1. Introduction

For clinical applications, the composition of body fluids has become a commonly used standard for diagnosing various diseases or monitoring of drug responses. There are several kinds of body fluids, such as urine [1], blood plasma [2], serum [3], cerebrospinal fluid [4] etc. Among these samples, blood/serum measurements are the cornerstone of clinical testing; thus, numerous investigations into the analysis of blood serum composition exist.

Rodents are the most popular animal model for pre-clinical studies; hence, it is very important to get blood samples as few animals as possible, and to improve data evaluation. Currently, there are many common sites for blood collection in rodents, such as the tail vein (easy for catheterization), retro-orbital sinus, facial vein, saphenous vein, heart or the inferior thoracic vena cava (ITVC) [5,6]. For terminal stage studies, blood collection sites from the heart or ITVC are preferred, due to the good quality volume of blood from animals. For the collection of multiple blood samples over a short period of time, the approaches of retro-orbital sinus, tail vein, or saphenous vein are appropriate. For the approach of retro-orbital sinus, the operator should be well trained, and the animal needs to be anaesthetized, or else this simple operation could seriously hurt the animal, resulting in, for instance, blindness [7,8]. Furthermore, it should be noted that anesthetics could alter the biochemical and hematological composition [9]. For tail vein collection, the operator should also be well trained, since some researchers just cut off the tail [10,11], which could seriously injure the animal, and the blood might be obtained from both vein and artery. Among these methods, the blood collection approaches-lateral saphenous vein/tail vein catheterization are relatively quick ways of collecting blood from all strains of rodents. Furthermore, the animal does not need to be anesthetized, but just needs slight restraining by hand. Thus, it is proposed that this is the best way of collecting multiple blood samples.

The protein compositions in the blood serum hold a wealth of information about the health status of patients. Furthermore, there is an increasing tendency towards studying the composition of small molecules, such as metabolites, due to the fact that their levels can be significantly influenced by many diseases, the administration of drugs, or by toxins [12,13]. Thus, blood measurements of metabolites have a wide range of applications [14] using various technologies, such as proton nuclear magnetic resonance spectroscopy (^1^H-NMR) [15], mass spectroscopy (MS) [16] and high-performance liquid chromatography (HPLC) [17], etc. Among these methods, ^1^H-NMR is the most often utilized to provide chemical and structural information of biological molecules [18] in a sample without any damage. The ^1^H-NMR spectra of blood serum are dominated by broad resonances from proteins and lipoproteins decorated by sharper resonances from small molecules. Aside from lipids, the dominant small molecule in the ^1^H-NMR spectrum of serum/plasma is glucose [19]. Furthermore, a number of amino acids and some organic acids are routinely detected, such as alanine, glutamine, leucine and histidine, lactate, citrate and succinate, etc. Concentrations of these metabolites are influenced by the brain state of the animal and the approach used of blood collection from the animal.

Thus, metabolomics studies of different blood collection approaches were investigated in the current study, i.e., different bleeding sites: saphenous vein/tail vein/ITVC; different brain states: awake/anesthesia; and different anesthetics: sevoflurane/pentobarbital. The chemical compositions of the serum from these different kinds of approaches were compared. This study verified that acute anesthesia could have an effect on blood compositions and that the bleeding site is also an influencing factor, especially for ITVC. Furthermore, this study provided efficient and convenient approaches for collecting multiple blood samples in a short period of time from awake animals.

## 2. Materials and Methods

### 2.1. Animals

The experimental protocols were approved by the animal care and use committee in Wuhan Institute of Physics and Mathematics, the Chinese Academy of Sciences. All male rats (*n* = 9; 8 weeks old) were ordered from VITAL RIVER (Beijing, China) and kept in SPF (Specific pathogen Free) animal residence (Wuhan, China). Rats were housed in plastic cages in a climate-controlled room with 12 h of light-dark illumination cycle at 25 ± 1 °C and relative 50 ± 10% humidity. During the experiment, all rats were allowed free access to laboratory standard food and water. To minimize stress on the day of the experiment, animals were weighed and handled daily for a week, including mildly touching the skin/hair, catching the animal, and holding the animals in the hand for about one minute. Operations of sample preparation can be found in the Appendix A.

### 2.2. Blood Collection

In order to compare the efficiency of blood collection methods from different bleeding sites, the brain states, and the anesthetized states under different anesthetics, six groups of blood samples were collected from the same animal: two from the saphenous vein under awake state (B_SV0_: *n* = 9; B_SV10_: *n* = 8); three from the tail vein under awake (B_TVA_: *n* = 6) and different anesthesia states (B_TVS_: *n* = 9 and pentobarbital-sample B_TVP_: *n* = 7); and the last one from the inferior thoracic vena cava (ITVC) under anesthetized state (pentobarbital sodium, B_ITVC_: *n* = 8). The whole experimental procedure and operation methods are illustrated in Figure 1. One animal died during the anesthesia procedure, and the blood collection operation was ceased after three attempts.

For saphenous vein (Blood samples: B_SV0_ and B_SV10_): The rat was first restrained by hand, and the hair on the tarsal joint was shaved. Then the hind limb was extended straight prior to blood collection, and the skin was smeared with Vaseline to avoid the blood spreading onto the skin and to facilitate the formation of blood clots. Using a fine 23 G needle, the first puncture was performed on the saphenous vein to collect the blood sample. Normally once is enough for bleeding, and the puncture times should not exceed three in one attempt, in line with animal care protocols. The bleeding was stopped by pressing a gauze or tissue on the puncture site. The animal was return to the home cage after the bleeding was totally stanched. The steps of blood collection are shown in Appendix A.

For tail vein under awake state (Blood sample: B_TVA_): A plastic animal holder and a specially designed syringe were needed for this approach. The syringe (1 mL) was connected to a fine 23G needle through a short length (~20 cm) of PE50 (O.D. 0.97 × I.D. 0.58 mm/L1.0m). At first, the awake animal was restrained in the plastic animal holder. Then, the tail vein catheterization was completed in the lateral tail vein with the special syringe (requiring more practice to achieve skilled operation) and the sample (~200 µL) was collected by withdrawing the blood with the syringe. The steps of the blood collection are illustrated in Appendix A.

For tail vein under anesthesia state (Blood samples: B_TVS_ and B_TVP_): Two different anesthetics were utilized in the current study: A: Sevoflurane (3–4%); B: 1% Pentobarbital (0.7 mL/100 g). In this step, the rats did not need to be restrained by the animal holder, and the level of anesthesia was verified by the loss of righting reflex, such as lack of withdrawal response to a foot pinch. Blood samples were directly collected from the lateral tail vein. The detailed steps are demonstrated in Appendix A.

For ITVC (Blood sample: B_ITVC_): At this point, a terminal procedure yielding maximal blood volume was performed. After completing the blood collection from the lateral tail vein in the anesthetized rat with pentobarbital, the rat chest was immediately opened to collect blood from the ITVC. At the end, ~200 μL blood sample was collected for further analysis.

Detailed information about the materials, appliance, bleeding rates and blood volume for different bleeding sites and brain states are illustrated in Table 1.

### 2.3. Sample Preparation

The collected blood samples were immediately centrifuged at 6000× *g* for 10 min at 4 °C, and the supernatant serum withdrawn by pipette and temporally stored on ice. After all the samples had been collected, the ice-cold serum (50 μL) was transferred to a 5 mm NMR tube, and mixed with 50 μL D_2_O (contained 5 mM formate) and 400 μL phosphate buffer (0.2 M Na_2_HPO_4_/NaH_2_PO_4_, pH 7.2). The samples were uniformity mixed by vortex and kept at −20 °C for further NMR analysis.

### 2.4. H-NMR Detection

To detect the small molecular weight metabolites, ^1^H-NMR spectra of the serum samples were obtained with Carr–Purcell–Meiboom–Gill (CPMG) pulse sequence in a Bruker AVANCE III 600 MHz CryoProbes NMR spectrometer (Bruker, Rheinstetten, Germany). The acquisition parameters were set as following: size of FID: 32 k; number of scans: 256; number of dummy scans: 4; spectral width: 20 ppm; 90° pulse length: 14.2 μs; spin-echo delay: 350 μs, number of loops: 80; and relaxation delay: 3.4 s.

### 2.5. NMR Spectra Processing

All NMR spectral data were analyzed with the commercial software Topspin 2.1 (Bruker Biospin, GmbH, Rheinstetten, Germany) and a home-made software *NMRSpec* [20] in MATLAB (R2018b, Mathworks Inc. 2018,) (Freely available from the author upon request: jie.wang@wipm.ac.cn).

All the FID signals of ^1^H-NMR spectra were converted by adding the exponential window function with a width increasing factor of 1Hz before the Fourier transformation (Topspin). Then the phase and baseline correction were performed manually in Topspin, and the chemical signals were calibrated with the inner standard-formate signal.

Furthermore, the NMR spectra data were imported to *NMRSpec* for peak alignment and integration. Then, continuous even spectral bucketing (0.004 ppm) and the areas of the whole peaks [20] in all spectra were automatically integrated in *NMRSpec*, and all bucketed spectra data and the peak areas were normalized using the probabilistic quotient normalization (PQN) method which was implemented in MATLAB [21] before comparing the total concentration differences.

Due to the overlapped signals in the ^1^H-NMR spectra, the relative concentrations of these metabolites were calculated based on the following procedures: the average chemical related peak area in the NMR spectra of S_SV0_ samples was set as the reference ‘1’. The areas of the same peaks in all samples were normalized using this reference, and the relative chemical concentration was calculated by averaging the normalized peak areas in the same locations of the NMR spectra in the same group.

### 2.6. Statistical Analysis

In order to initially compare the differences of the metabolites in these different types of blood collection approaches, the correlation of the metabolites in the NMR spectra were analyzed using the PQN normalized areas of the peaks in the NMR spectra.

Then, the normalized data was imported into the SIMCA-p + software package (v11.0, Umetrics, Malmö, Sweden) for multivariate statistical analysis. With the adoption of UV standardization of pre-processing method, Principal component analysis (PCA) is mainly used for the observation of sample clustering of the whole situation and the existence of outliers.

Then, the difference between these three different kinds of serum samples were analyzed with the help of PLS-DA method (Partial Least Squares Discriminant Analysis). The PLS-DA method is a classification algorithm based on partial least squares algorithm. Its function is to use the mathematical model established by X to predict the classification of unknown samples in Y, at the same time maximizing the separation of the two groups, which is helpful to find out the metabolites that contribute to the classification. The significant varying metabolites were extracted from OPLS-DA correlation coefficient color coded loading plots.

## 3. Results and Discussion

### 3.1. Blood Collection Methods

Many blood collection methods have been reported [5,22]. As stated in the introduction, most of these methods have adverse effects such as tissue damage and contamination from the glands [8,23] in awake animals. In order to avoid these problems, the animal could be anesthetized, which could influence the metabolic components in the serum. Furthermore, most of these methods could not be utilized with multiple sample collections from the same animal under an awake state. The saphenous vein and tail vein blood collection methods have the minimum adverse effects on the animals; this approach could be selected as the best representative of peripheral blood samples for potential multiple samples collection in the same animal.

At first, three animals were appropriately anesthetized with sevoflurane during the second blood collection. The rats showed a reduction in heart rate and blood pressure following sevoflurane anesthesia. The blood vessels contracted and the rate of bleeding significantly decreased (almost no bleeding) after puncture. Thus, the blood samples from the saphenous vein were only collected under the awake state.

To demonstrate the effect of different brain states and body sites on the blood sampling procedure in rats (*n* = 9), various blood collection methods were implemented under different conditions in the current study, such as blood collections from different sites-saphenous vein/tail vein/ITVC, different brain states-awake/anesthesia, different anesthetics-sevoflurane/pentobarbital.

### 3.2. Variation of ^1^H-NMR Spectra of Blood under Different Blood Collection Methods

An example of ^1^H–NMR spectra of the serum is shown in Figure 2A and Figure 3-Blood. The peak assignments and chemical shifts of the signals in ^1^H-NMR are illustrated in Table 2. The nuclear magnetic signals of the metabolite were attributed to the laboratory data based on the two-dimensional spectrum COSY, TOCSY, JRES, HSQC and HMBC, as well as the related literature [12,13,15,18,24,25,26] and public databases (HMDB). These spectra mainly include the NMR signals of organic acid, amino fatty acid and creatinine as well as other metabolites such as choline metabolite ethanolamine purine and pyrimidine metabolites.

In order to evaluate the stability of these different blood collection methods, the average and the standard error of the mean (SEM) of the ^1^H-NMR spectrum in every group were calculated point by point after the spectral peak alignment was achieved (Figure 2B) [27]. The major metabolites in the blood sample are more stable with the blood collection methods in the awake animal from the saphenous vein (0 day-S_SV0_ or 10 days later-S_SV10_) or tail vein (S_TVs_ or S_TVA_). The anesthesia could contribute to the variation of the components especially for pentobarbital (S_TVP_). Furthermore, the bleeding site of ITVC could be used to obtain blood samples as often as necessary; however, the metabolic components are too varied, especially for glucose (S_ITVC_).

To initially evaluate the differences of the serum samples under various blood collection methods, differences of the ^1^H-NMR spectra for blood samples under different conditions were calculated, and are illustrated (Figure 3A–F), such as blood collection sites, brain states and anesthetics. This figure shows the tendency of the small metabolites and lipids among different samples. The major metabolic components in the blood sample are almost the same as those in the same brain state and blood collection site in the close period (Figure 3A, 10 days’ difference, except for lactate and lipids), even for the different blood collection sites (Figure 3B). The anesthesia state could influence the compositions of the blood samples (Figure 3D,E), and different anesthetics have different effects (Figure 3(F)). The blood collection method from the ITVC had the most significant influence on the metabolites in the blood sample, especially for glucose (Figure 3C). Without the involvement of standard deviation, it was not possible to illustrate the statistical difference among the samples; thus, it was very important to do the statistical analysis in order to describe the difference and estimate the effect of anesthesia on the blood composition.

### 3.3. Correlations of ^1^H-NMR Spectra of Blood Samples from Different Blood Collection Methods

To further explore the effects of different brain states, anesthetics and blood collection sites, correlations of ^1^H-NMR spectra of different blood samples were compared. Here, the average PQN normalized peak areas in the same group were utilized for calculation.

The correlations of various blood samples were calculated and illustrated (Figure 4A–F), such as same site (different periods), different sites (Saphenous vein/tail vein in awake state or Tail vein/ITVC in pentobarbital induced anesthesia state), awake and anesthesia state (sevoflurane/pentobarbital) or different anesthetics (sevoflurane/pentobarbital).

Comparing the blood samples from the saphenous vein under awake state in different periods, the major metabolic components of the blood samples were almost similar, except for lipids, which is far from the central line (*y = x*, Figure 4A, *r* = 0.9884,). Furthermore, lipids are also the major different component in different blood collection sites (saphenous vein vs. tail vein, Figure 4B) under awake state, however, the other small metabolites were almost similar (*R* = 0.9555). It was more stable under the anesthesia state induced by pentobarbital (Tail vein vs. ITVC, Figure 3(C)), especially for lipids. Under different anesthetics, the major metabolic components were varied (Figure 3(D–F)), especially for pentobarbital (*R*: 0.9087 for pentobarbital vs. 0.9549 for sevoflurane). Pentobarbital is a liquid anesthetic; thus, it probably changes the components of the blood sample more significantly. In order to check the contribution of the metabolites to the discrimination, further statistical analyses were implemented in the next section.

### 3.4. Principle Component Analysis

To determine whether it was possible to distinguish the samples from different blood collection methods and screen the possible outliers, the unsupervised pattern recognition method PCA was performed on the PQN normalized NMR data. The normalized continuous even spectral bucketing data were utilized for analysis, and PCA was used to reduce the dimensions of the variables by dropping the unnecessary data. The principle components were calculated with the combination of the major variables.

For all the samples, the top three principle components were calculated, which made 43.7%, 19.5% and 8.7% contribution to the total component, respectively. These three principle components made up a total of 71.9% of the variance and may play major roles. The loading plot of the samples with these three major components is illustrated in Figure 5. It should be noted that there was a separation trend for some samples, such as B_ITVC_ v.s. B_SV0_ or B_SV10_ or B_TVS_, B_TVA_ v.s. B_TVP_, etc. Some samples were difficult to distinguish, such as B_SV0_ v.s. B_SV10_ or B_TVA_. Most of the samples were overlapped in the 3D-space; thus, the PCA analysis for two difficult kinds of samples were implemented in the next step.

At the end, the first two components (PC1 and PC2) were calculated for six pairs of two different kinds of samples, which are shown in Figure 6. The total contributions of the first two components were higher than 50% of the variance of all variables and played major roles in the discrimination analysis. The loading plots of PCA results indicated that there were no outliers among the serum samples obtained from different approaches, as demonstrated by the clustering observed in the PCA results (Figure 6A–E). There was a separation tendency for group discrimination in every comparison. Thus, these preliminary results indicate that there could be some different metabolites in various blood samples, even from the same site on different days.

### 3.5. OPLS-DA Analysis

In order to specifically screen the different characteristics of serum collected using different methods, OPLS-DA models were constructed for further analysis. Parameters of R^2^X and Q^2^ are the main parameters for the model validation, R^2^X is used to explain the difference between the models, and Q^2^ reflects the ability of the prediction of the models. Results of R^2^X and Q^2^ for six different discriminate classification models are shown in Table 3, including the same bleeding site in different periods (S_SV0_ vs. S_SV10_), different bleeding sites under awake/anesthesia state (S_SV10_ vs. S_TVA_ and S_TVP_ vs. S_ITVC_), different anesthesia states induced by sevoflurane/pentobarbital (S_SV10_ vs. S_TVS_; S_SV10_ vs. S_TVP_ and S_TVS_ vs. S_TVP_).

The results of these six different pattern recognition analyses are represented as score scatter plots (Figure 7A1–F1), which show the inherent clustering trends of the samples. The coefficient-coded loading plots established by MATLAB script were employed to identify the significant contributing metabolites among the serum samples from the saphenous vein and inferior thoracic vena cava (Figure 7A2–F2).

The loading plots indicate that there were significant differences for the metabolites in these six pairs of blood samples. For the awake state, the small molecules were almost similar from the same site in different periods or different sites in the same periods. However, the lipids and lactate were different in these two comparisons (Figure 7A2,B2). For the comparison of different bleeding sites under the anesthesia state, the lipids are more stable, but the glucose was increased (Figure 7C2). Anesthetics could change both small molecules and lipids (Figure 7D2,E2), such as increasing glucose, alanine, glycerol and arginine, and decreasing lactate, lipid and glutamine, especially pentobarbital (Figure 7E2). However, different anesthetics have different effects (Figure 7F2). Among these metabolites, it is noticeable that the metabolites of glucose, lipid and lactic acid were the most significant components. Thus, these metabolites were extracted for comparison in the next section.

### 3.6. Metabolites in Different Kinds of Blood Samples

According to the results of OPLS-DA, the most significant different metabolites were lipids, lactate and glucose. The relative concentrations of these metabolites were calculated (Figure 8).

Glucose was almost similar in the serum under awake condition from different sites or at close period (10 days’ difference). It was significantly increased under the anesthesia state [28], even for a very short period in the current study. Comparing both anesthetics, pentobarbital influenced glucose more significantly, especially for the ITVC group, which might be caused by a longer time under anesthesia. Former studies have shown that sevoflurane anesthesia could inhibit pancreatic β-cells secreting insulin, which may be caused by the continuous activation of adenosine triphosphate–sensitive potassium channels in β-cells [29]. Insulin can induce the synthesis of glucokinase and promote glycolysis in the liver. When the concentration of insulin decreases, the hepatic glucose homeostasis shifts from glycogenolysis/gluconeogenesis to glycogen synthesis via insulin signaling, which increases the concentration of glucose in blood [29]. In the meanwhile, there is an extensive inhibition of substrate oxidation in muscle mitochondria, which reduces glucose uptake. In fat tissue, a main effect of insulin is the suppression of lipolysis and the reduced release of non-esterified fatty acids. In addition, lipolysis and the concentration of fatty acid increase with the inhibition of insulin secretion.

Lipids and lactate were varied among different samples, even from the same site under different periods or different sites on the same day. Thus, the influential factors could be very complicated for these components, and it could be very difficult to verify the influence of a single factor in other studies.

The purpose of this study was to identify whether there was any difference in the samples obtained using different sampling ways, such as the same bleeding site in different periods, different bleeding site under awake/anesthesia state, or different anesthesia state induced by sevoflurane/pentobarbital. Among these various samples, the ITVC group showed the most changes in metabolites, and probably should not be used for the metabolic analysis in animal studies. Furthermore, Bernardi et al. also found that the total leucocytes, absolute neutrophil and lymphocyte counts were significantly higher in the samples collected from the peripheral sites than those samples collected from ITVC [30].

### 3.7. Characteristics of Blood Collection Methods

Comparing these six different kinds of serum samples, the metabolites were varied, even from the same method under different periods. Thus, we suggest that blood samples should be obtained on the same day using the appropriate blood collection method.

Among various blood collection methods, the bleeding site from the saphenous vein is the most convenient choice, even without any training. In the current study, there was no failed experiment; however, the total blood volumes were varied at different times. For the tail vein, the operator should be well trained, especially for awake animals. There was one failure in the anesthesia state (17 times in total) and two in the awake state (8 times in total). However, more blood samples could be obtained employing this method, even more than 0.5 mL per time, and the bleeding rate is much faster than with the saphenous vein method. Both methods could be used for multiple blood collection. For the method of ITVC, the terminal procedure yielded maximal blood volume; however, it was used only once. Furthermore, the changes of the metabolites using this method should be considered, as the blood components were different from the other site when the animal was under the same anesthesia condition (B_TVP_ vs. B_ITVC_).

## 4. Conclusions

The current study compared the metabolites in blood samples using various blood collection methods, including different bleeding sites, i.e., saphenous vein/tail vein/ITVC, awake/anesthetized rats, and sevoflurane/pentobarbital induced anesthesia. The metabolic components in the blood were influenced by the brain state of the animal, and anesthesia could significant increase the glucose concentration and decrease the lactate concentrations, especially from ITVC. Therefore, the choice of the most suitable sampling site should be selected according to the experimental requirements. From the perspective of animal welfare and multiple sampling, saphenous vein blood collection is a simpler, more convenient and appropriate method. Furthermore, the tail vein blood collection method is another suitable method for sufficient volume blood collection, but the operator needs to be skilled. Both of these two methods are suitable for multiple blood collections, pharmacokinetics and for studying the effects of drug interventions on animals.

## Figures and Tables

**Figure 1 molecules-24-02542-f001:**
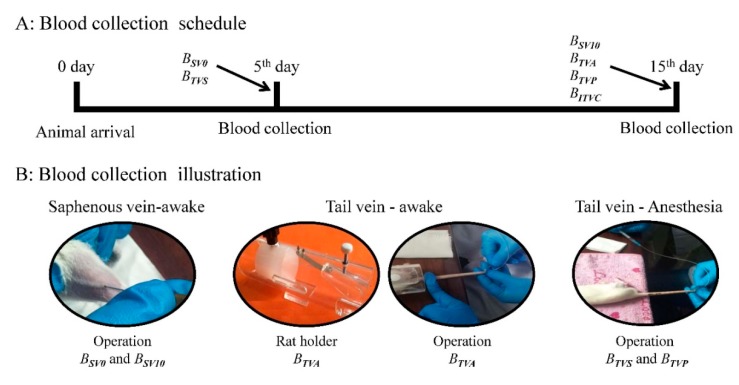
Flow chart of the whole experimental procedure (**A**) and demonstrations of the blood collection from the saphenous vein and the tail vein under awake/anesthesia states (**B**). Note: B: blood; SV0 or SV10: Blood collection from the saphenous vein after 0th or 10th day. TVA, TVS or TVP: Blood collection from the tail vein under awake or anesthesia state induced by sevoflurane or pentobarbital; ITVC: Blood collection from the inferior thoracic vena cava.

**Figure 2 molecules-24-02542-f002:**
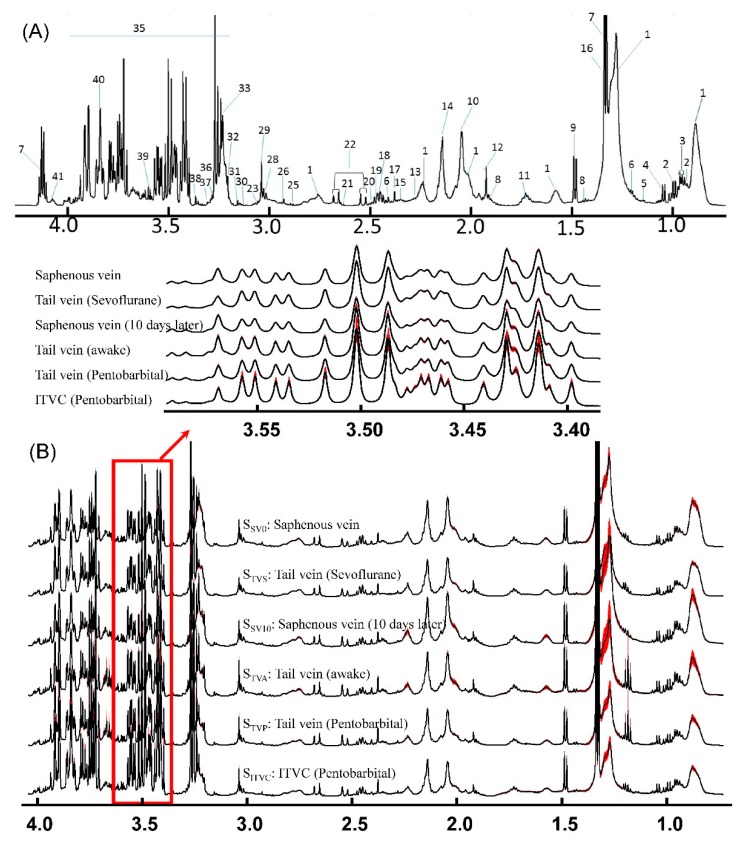
^1^H-NMR Spectra of blood samples. (**A**) Peak assignments of ^1^H-NMR spectroscopy of one random blood sample; (**B**) Averaged ^1^H-NMR spectra plus its’ SEM values point by point for various blood samples from different collection methods. Note: S: ^1^H-NMR signals of various blood samples; Subscript: Please see Figure 1; Labels in Figure 2A are demonstrated in Table 2.

**Figure 3 molecules-24-02542-f003:**
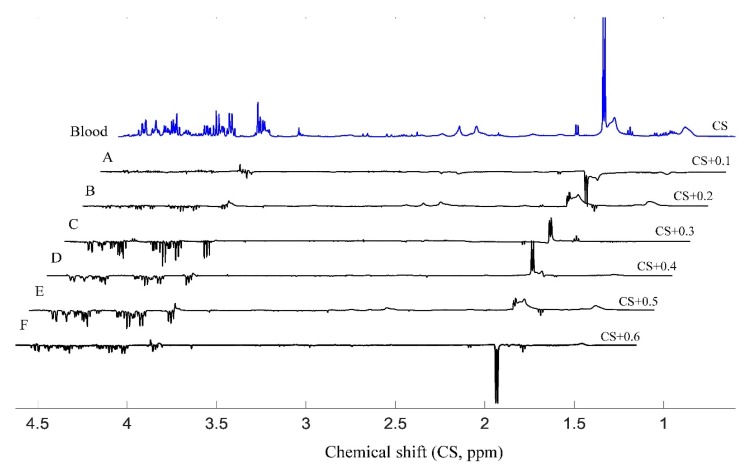
Differences of the average ^1^H–NMR spectra of blood samples from different approaches. A) Difference in blood samples from the same blood collection site (saphenous vein) under awake state in different periods (S_SV0_-S_SV10_); B): Effect of different blood collection sites under awake state (S_SV10_-S_TVA_); C): Effect of different blood collection sites in anesthesia state (S_TVP_-S_ITVC_); D): Effect of sevoflurane on blood composition (S_SV0_-S_TVS_); E): Effect of pentobarbital on blood composition (S_SV10_-S_TVP_); F): Effect of different anesthetics on blood compositions (S_TVS_-S_TVP_). *Note**: CS: chemical shift*.

**Figure 4 molecules-24-02542-f004:**
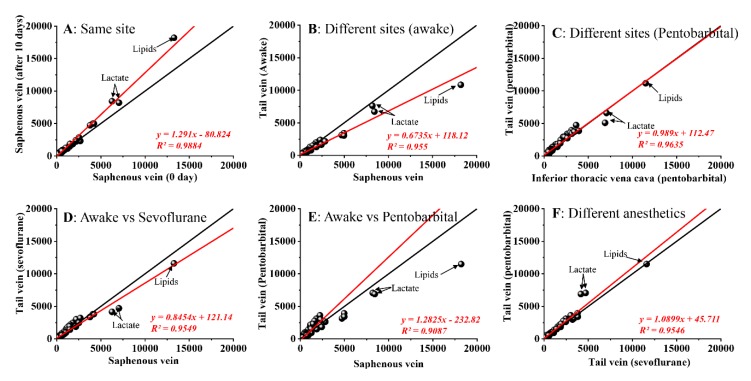
Correlations of different kinds of blood samples, including same site under different periods (**A**), different sites under various brain states (**B**,**C**), and different kinds of anesthetized states (**D**–**F**).

**Figure 5 molecules-24-02542-f005:**
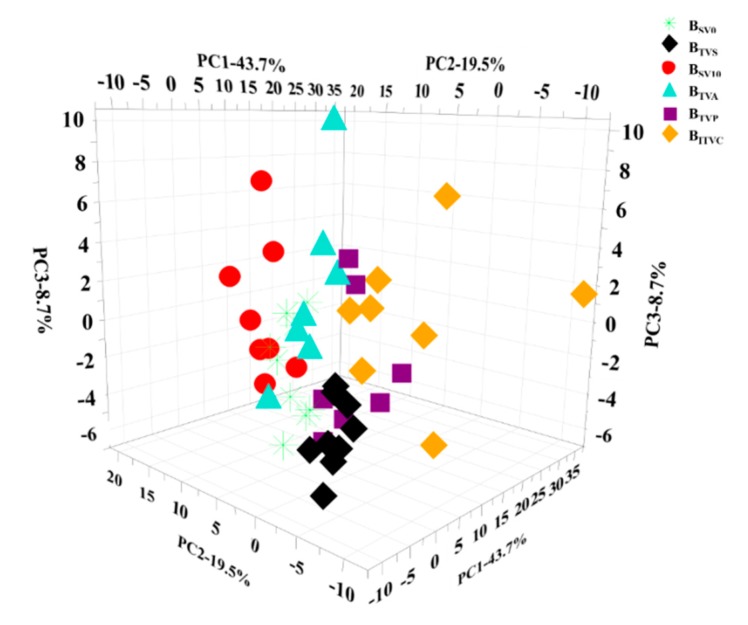
Loading plots of principal component analysis (PCA) of the NMR spectral data for all serum samples. Note: Every sample is represented by a unique pattern.

**Figure 6 molecules-24-02542-f006:**
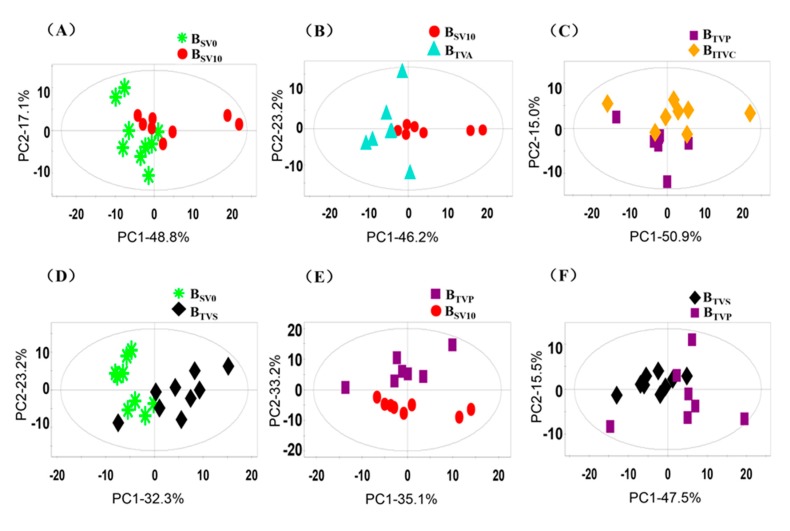
Loading plots of principal component analysis (PCA) of the NMR spectral data from various blood samples. Note: Every sample is represented by a unique pattern. Note: (**A**): B_SV0_ vs. B_SV10_; (**B**): B_SV10_ vs. B_TVA_; (**C**): B_IVP_ vs. B_ITVC_; (**D**): B_SV0_ vs. B_TVS_; (**E**): B_TVP_ vs. B_SV10_; (**F**): B_TVS_ vs. B_TVP_.

**Figure 7 molecules-24-02542-f007:**
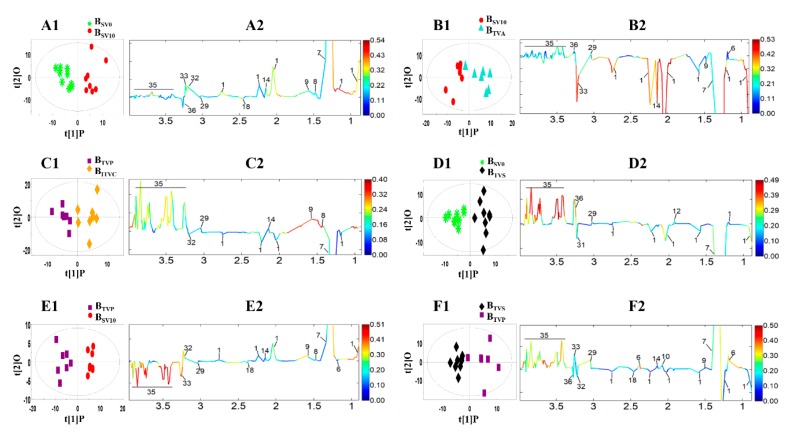
Score plots (**A****1**–**F1**) and coefficient-coded loadings plots (**A2**–**F2**) from the results of OPLS-DA derived from ^1^H-NMR spectra of the serum samples from six different comparisons *A:* S_SV0_ vs. S_SV10_; B: S_SV10_ vs. S_TVA_; C: S_TVP_ vs. S_ITVC_; D: S_SV10_ vs. S_TVS_; E: S_SV10_ vs. S_TVP_; F: S_TVS_ vs. S_TVP_. Note: The serial number of the metabolite signal is shown in Table 1.

**Figure 8 molecules-24-02542-f008:**
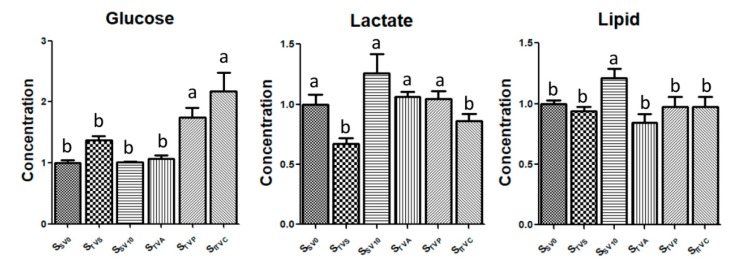
Relative concentrations of the lipids, lactate and glucose in six different serum samples. Note: Different letters represent significant differences at *p* < 0.05.

**Table 1 molecules-24-02542-t001:** The surgical materials for different kinds of blood sample collection approaches.

Bleeding Site	Materials	Appliance	Animal State	Bleeding Rate	Blood Volume
Saphenous vein	Needles (23G), Vaseline	Electric shaver	Awake	3.66 ± 0.72 μL/s	~400 μL
Tail vein	Needles (23G), Syringe (1 mL), PE50 tubing	Rat holder, Tweezer	Awake	8.50 ± 1.70 μL/s	~400 μL
Tail vein	Needles (23G), Syringe (1 mL), PE50 tubing (0.058 cm × 0.097 cm)	Tweezer	Anesthetic (sevoflurane/pentobarbital sodium)	8.50 ± 1.70 μL/s	~500 μL
Inferior thoracic vena cava	Syringe (2 mL)	Scissor	Anesthetic (pentobarbital sodium)		~2 mL, even more

**Table 2 molecules-24-02542-t002:** The NMR related information of the related proton signals in the small molecules in the serum samples.

Metabolites	Moieties	^1^H Shift(δ)	Peak Num	Structure
Lipid	CH_3_	0.891(t)	1	
	CH_3_CH_2_CH_2_	1.210(m)	
	(CH_2_)_n_	1.221(m)	
	CH_3_CH_2_(CH_2_)_n_	1.232(m)	
	CH_2_ CH_2_CO	1.590(m)	
	CH_2_C=C	2.018(m)	
	CH_2_COO	2.238(m)	
	C=CCH_2_C=C	2.742(m)	
	C=CCH_2_C=C	2.749(m)	
	C=CCH_2_C=C	2.761(m)	
Isoleucine	δCH_3_	0.943(t)	2	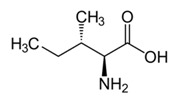
	βCH_3_	1.000(d)	
	Γ′CH_3_	1.008(d)	
	γCH_2_′	1.284(m)	
	γCH_2_	1.459(m)	
	βCH	1.961(m)	
Leucine	δCH_3_	0.955(d)	3	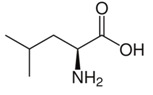
	δ′CH_3_	0.965(d)	
	δCH_3_	0.975(d)	
	γCH	1.691(m)	
	βCH_2_	1.707(m)	
	αCH	3.685(dd)	
	αCH	3.753(d)	
Valine	γ′CH_3_	0.988(d)	4	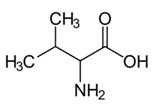
	CH_3_	1.020(d)	
	CH_3_	1.040(d)	
	γCH_3_	1.052(d)	
	βCH	2.285(m)	
	αCH	3.570(d)	
	αCH	3.617(d)	
Isobutyrate	CH_3_	1.361(d)	5	
3-hydroxybutyrate	γCH_3_	1.200(d)	6	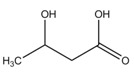
	αCH_2_	2.293(m)	
	αCH_2_	2.380(m)	
	βCH	4.131(m)	
Lactate	βCH_3_	1.341(d)	7	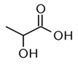
	αCH	4.108(q)	
Lysine	half δCH2	1.434(m)	8	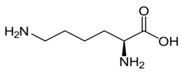
	half δCH2	1.689(m)	
	γCH2	1.719(m)	
	half βCH2	1.886(m)	
	half βCH2	1.897(m)	
	εCH2	3.031(t)	
	αCH	3.767(t)	
Alanine	βCH_3_	1.480(d)	9	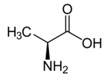
	γCH_2_	1.492(m)	
	αCH	3.783(q)	
NAG	CH_3_	2.041(s)	10	
Arginine	γCH_2_	1.681(m)	11	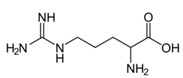
	γCH_2_	1.730(m)	
	βCH_2_	1.926(m)	
	δCH_2_	3.257(t)	
	αCH	3.774(m)	
Acetate	CH_3_	1.914(s)	12	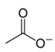
Acetoacetate	CH3	2.273(s)	13	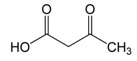
	CH2	3.441(s)	
OAG	CH_3_	2.140(s)	14	
Glutamine	γCH_2_	2.411(m)	15	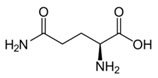
	γCH_2_	2.465(m)	
	αCH_2_	3.677(t)	
Threonine	γCH3	1.329(d)	16	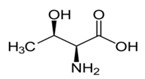
	αCH	3.487(d)	
	αCH	3.593(d)	
Pyruvate	CH_2_	2.318(s)	17	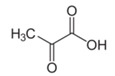
	CH_3_	2.372(s)	
Glutamate	βCH2	2.077(m)	18	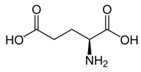
	γCH2	2.351(m)	
	αCH	3.786(t)	
Succinate	CH_2_	2.395(s)	19	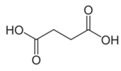
2-ketoglutarate	αCH_2_	2.437(t)	20	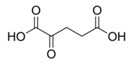
Methionine	αCH_2_	3.858(m)	21	
	βCH_2_	2.166(t)	
	γCH_2_	2.657(t)	
	δCH_2_	2.142(s)	
Citrate	half CH_2_	2.523(d)	22	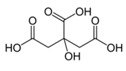
	half CH_2_′	2.657(d)	
Tyrosine	half βCH2	3.058(dd)	23	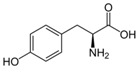
	half βCH2	3.158(dd)	
	β′CH2	3.199(dd)	
	αCH	3.951(dd)	
Asparagine	half βCH_2_	2.836(dd)	25	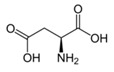
	half βCH_2_	2.941(dd)	
	βCH_2_′	2.948(dd)	
	αCH	3.997(dd)	
Dimethylglycine	N-CH_3_	2.930(s)	26	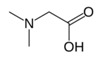
	CH_2_	3.723(s)	
2-ketoisovalerate	γCH3	1.111(d)	27	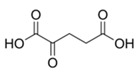
	βCH	3.020(m)	
Creatine	CH_3_	3.040(s)	28	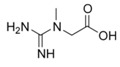
	CH_2_	3.938(s)		
Creatinine	CH_3_	3.051(s)	29	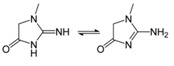
	CH_2_	4.066(s0	
Phenylalanine	βCH_2_	3.119(dd)	30	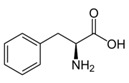
	B′CH_2_	3.260(dd)	
	αCH	3.962(dd)	
	αCH	3.991(dd)	
Choline	N(CH_3_)_3_	3.208(s)	31	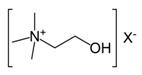
	NCH_2_	3.657(m)	
	OCH	4.072(m)	
Phosphocholine	N(CH_3_)_3_	3.218(s)	32	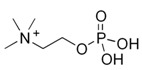
	N CH_2_	3.585(m)	
	OCH_2_	4.142(m)	
Glycerophosphochline	N(CH_3_)_3_	3.233(s)	33	
α-glucose	H4	3.429(t)	35	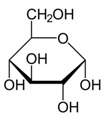
	H2	3.542(dd)	
	H3	3.708(t)	
	half CH2-C6	3.732(dd)	
	H5	3.822(dd)	
	H6	3.840(dd)	
β-glucose	H_2_	3.242(dd)	35	
	H_4_	3.398(t)		
	H_5_	3.468(dd)		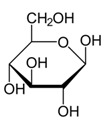
	H_3_	3.503(t)	
	H_6_	3.743(dd)	
	H_6_′	3.898(dd)	
Betaine	N(CH_3_)_3_	3.271(s)	36	
	OCH_2_	3.915(s)	
Taurine	CH_2_SO_3_	3.271(t)	37	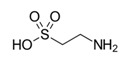
	CH_2_SO_3_	3.414(t)	
Scyllo-inositol	CHOH	3.355(s)	38	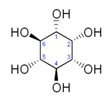
Glycine	CH_2_	3.558(s)	39	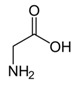
Glycerol	half CH_2_	3.552(dd)	40	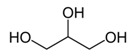
	half CH_2_	3.649(dd)	
	CH	3.795(m)	
Triglycerides	CH_2_O	4.072(m)	41	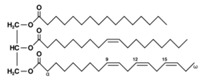

Note: The peak assignments were identified based on former publications [12,13,15,18,24,25,26].

**Table 3 molecules-24-02542-t003:** Statistical parameters of OPLS-DA analysis for different samples.

Statistical Parameter	S_SV0_ vs. S_SV10_	S_SV10_ vs. S_TVA_	S_TVP_ vs. S_ITVC_	S_SV0_ vs. S_TVS_	S_SV10_ vs. S_TVP_	S_TVS_ vs. S_TVP_
R^2^X	0.601	0.613	0.645	0.459	0.408	0.589
Q^2^	0.625	0.522	0.672	0.672	0.759	0.418

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
