# Peer review of "NMR Based Metabolomics Comparison of Different Blood Sampling Techniques in Awake and Anesthetized Rats"

_molecules, 2019, doi:10.3390/molecules24142542_

Round 1
Reviewer 1 Report
The authors have significantly added new data compared to the 1st draft for strengthening the content, but the overall conclusion – both in writing and content – is weak; the metabolite changes in lactate and glucose is not ‘new’.
Since the authors adopt various data analyses (spectra, PQN, OPLS, quantifications…) to identify the three metabolite variants (lac, glc and lip), the authors can expand the discussion/conclusion by stating clearly how these specific findings correlate to the statement lines 602-607. This is missing throughout the manuscript!
-fig 3 what is CS (control spectrum??)
-fig 5 PCA: they are score plots not loading plots. Why carry out a separate PCA analysis for the different groups? since PCA is an unsupervised analysis, it gives more info of the model study if ALL the groups are compared in PCA. (and the loading ‘may’ also identify the metabolite variances)
-fig 7: in text it stated ‘absolute concentration’, but in the figure caption it states ‘relative’ which is correct? If ‘absolute’, how did you quantified the metabolite (what chemical shift, internal or external referencing??). If ‘relative’, how did you carry out quantification? What are they relative to? State the level of difference p-values of ‘a’ and ‘b’
-typos: line 128; ref 23
Author Response
Response to reviewer 1:
Question 1: The authors have significantly added new data compared to the 1st draft for strengthening the content, but the overall conclusion – both in writing and content – is weak; the metabolite changes in lactate and glucose is not ‘new’.
Since the authors adopt various data analyses (spectra, PQN, OPLS, quantifications…) to identify the three metabolite variants (lac, glc and lip), the authors can expand the discussion/conclusion by stating clearly how these specific findings correlate to the statement lines 602-607. This is missing throughout the manuscript!
Answer: Thank you so much for your careful review. We have carefully revised our manuscript and asked a native English speaker Prof. Anne Manyande (University of West London) to complete the proofreading. We also add a paragraph for expanding the discussion of the specific findings in the manuscript (L368-383).
Question 2: -fig 3 what is CS (control spectrum??)
Answer: The information about CS (chemical shift) has been added in the revised manuscript (L268-269).
Question 3: -fig 5 PCA: they are score plots not loading plots. Why carry out a separate PCA analysis for the different groups? since PCA is an unsupervised analysis, it gives more info of the model study if ALL the groups are compared in PCA. (and the loading ‘may’ also identify the metabolite variances)
Answer: That is a good question. Results of PCA for all samples were added in the revised manuscript (L277-284).
Question 4: -fig 7: in text it stated ‘absolute concentration’, but in the figure caption it states ‘relative’ which is correct? If ‘absolute’, how did you quantified the metabolite (what chemical shift, internal or external referencing??). If ‘relative’, how did you carry out quantification? What are they relative to? State the level of difference p-values of ‘a’ and ‘b’
Answer: Thank you for your careful review. It was relative concentration. The description of the method has been added in the revised manuscript (L163-168 &L366).
Question 5: -typos: line 128; ref 23
Answer: They have been corrected in the revised manuscript.
Reviewer 2 Report
The manuscript by Du and Li et al. investigates perturbations in metabolic profiles of blood samples base on the different sampling techniques and type of anesthesia. This study is of interest, and the Authors substantially improved the impact of this manuscript (comparing to the initial submission). In particular, incorporation of new sampling groups and new data analysis enrich this study. However, manuscript suffers from lack of consistency in data presentation, and poor English. I recommend, addressing following issues prior to publication:
Major comments:
1. The manuscript is written using very poor English. Even, though, I am not a native English speaker, I could easily find numerous grammar mistakes. These needs to be extensively corrected. Otherwise, it is hard to follow the description and interpretation of the data. Here are just few examples to illustrate problems with language in this manuscript:
a. Page 1, line 41-42: “thus there were substantial investigations exists concerning composition analysis of blood serum”. Example correction: ”thus, substantial investigations concerning analysis of blood serum composition exist”
b. Page 2, line 114-115: “Furthermore, the small molecule composition, such as metabolites, also gained increased attention, due to the metabolites is seriously influenced by many diseases,..”. Example correction: “Furthermore, there is increasing attention towards studying composition of small molecules, such as metabolites, due to the fact that their levels can be significantly influenced by many diseases,…”
c. Page 3, line 212-213: “This is the terminal point of the whole experiment, thus the blood volumes could be obtained as much as we can”. Example correction: “At this point, a terminal procedure yielding maximal blood volume was performed.”
d. Page 12, line 402: “almost similar” should be “almost the same”.
Etc. I do not list all the mistakes because there is too many of them. Above I present just examples.
2. Why number of animals has been changed from 15 to 8? Page 2, line 137. Is that because not all animal underwent the same procedures and only 8 are paired? Or there was some other reason?
3. I appreciate, Authors effort to explain sampling order using the flow chart. However, it is still not very easy to understand what the exact experimental setup was. I understand that, sample order is on the timeline, and collection method is shown with the figures. But, at first view all of this is very confusing. E.g. which sample was collected using “Rat holder-awake”? Or what is the difference between Sample I and Sample III collected on day 0 and day 15 (is it just the time between the collections?). Perhaps, it would be more useful to align, figures vertically under the day of collection, so it would be straightforward what procedures were done on which day. Moreover, the manuscript lack consistency with regard to naming the sample groups. For example, in PCA plots (Figure 5) same names are used as in the flow chart (e.g. Sample III). However, in Figures 2,3,4,6 and 7 different naming patter was applied (e.g. SSv0 and STvs). Authors should decide on one naming pattern and apply it consistently through the text.
4. Plasma is the liquid, cell free, part of blood. Therefore, in order to obtain blood plasma, the use of anticoagulants, such as EDTA, is required. When blood is allowed to clot than and centrifuged, the remaining fluid is serum. Serum and plasma have similar but not identical metabolic composition. Please clarify, if there were anticoagulants used in this study, and if not change the wording, from “plasma” to “serum”.
5. There appear to be some erroneous assignments. For example: Peak 10 is assigned to citrulline, but it looks more like a lipid signal (break resonance). It is more likely the CH2 group of VLDL lipid. Example assignment of could be find in these papers:
a. https://www.ncbi.nlm.nih.gov/pubmed/25213261
b. https://www.ncbi.nlm.nih.gov/pubmed/7762816
Also, how is 3-hydroxyisobutyrate (peak 45) so high at 3.5 ppm, while the doublet at 1 ppm is almost absent? I recommend, verifying the metabolite assignments.
6. Figure 2B. It is supposed to be averaged NMR spectrum with SEM value. Where are SEM values? Also, in the text (Page 11, line 378-384) there is no reference to the figure. Authors claim that the variability of metabolites is different in different sampling methods, but there is no data to back it up.
7. Figure 4. This figure is very useful, but it could be improved. In particular, the regression line calculated from the data (the one that result in R2 values presented in the plots) should be plotted in addition to the y=x line.
8. Figure 5, the number of samples in groups is very suspiciously. It is stated in the text, that there were 8 animals in total (Page 2, line 137) and samples were collected from these 8 animals at different times, using different methods. However, upon investigation of Figure 5 these are the numbers of samples:
a. Fig5A: Sample I n= 9, Sample III n=8
b. Fig5B: Sample III n=7, Sample IV n=6
c. Fig5C: Sample V n=6, Sample VI n= 8
d. Fig5D: Sample I n=9, Sample II n=9
e. Fig5E: Sample V n=7, Sample III n =8,
f. Fig5F: Sample II n=9, Sample V n=7
My calculations for some of n<8 numbers may not be accurate due to points overlap and actually all 8 samples were used for plotting. However, there is no excuse for having more points than are supposed to be in the group. Please clarify, what happened here.
Minor comments:
1. Page 1, line 40, cell culture media is not a body fluid.
2. Page 4, line 244, change: “The mixtures” to “The samples”.
3. Page 5, line 316, change: “A sample” to “An example”.
4. Please explain how the concentrations for quantified metabolites (Figure 7) were obtained.
Author Response
Response to reviewer 2:
The manuscript by Du and Li et al. investigates perturbations in metabolic profiles of blood samples base on the different sampling techniques and type of anesthesia. This study is of interest, and the Authors substantially improved the impact of this manuscript (comparing to the initial submission). In particular, incorporation of new sampling groups and new data analysis enrich this study. However, manuscript suffers from lack of consistency in data presentation, and poor English. I recommend, addressing following issues prior to publication:
Answer: Thank you so much for your carefully review and professional criticism. We have carefully revise our manuscript, and also asked a native English speaker Prof. Anne Manyande (University of West London) for the proofreading.
Question 1: The manuscript is written using very poor English. Even, though, I am not a native English speaker, I could easily find numerous grammar mistakes. These needs to be extensively corrected. Otherwise, it is hard to follow the description and interpretation of the data. Here are just few examples to illustrate problems with language in this manuscript:
a. Page 1, line 41-42: “thus there were substantial investigations exists concerning composition analysis of blood serum”. Example correction: ”thus, substantial investigations concerning analysis of blood serum composition exist”
b. Page 2, line 114-115: “Furthermore, the small molecule composition, such as metabolites, also gained increased attention, due to the metabolites is seriously influenced by many diseases,..”. Example correction: “Furthermore, there is increasing attention towards studying composition of small molecules, such as metabolites, due to the fact that their levels can be significantly influenced by many diseases,…”
c. Page 3, line 212-213: “This is the terminal point of the whole experiment, thus the blood volumes could be obtained as much as we can”. Example correction: “At this point, a terminal procedure yielding maximal blood volume was performed.”
d. Page 12, line 402: “almost similar” should be “almost the same”.
Etc. I do not list all the mistakes because there is too many of them. Above I present just examples.
Answer: Thank you so much for your careful review. We have revised our manuscript carefully, and also asked a native English speaker Anne Manyande (University of West London) to completed the proofreading.
2. Why number of animals has been changed from 15 to 8? Page 2, line 137. Is that because not all animal underwent the same procedures and only 8 are paired? Or there was some other reason?
Answer: Thank you for your careful review. Comparing with the 1st version, the current experimental data was totally new. The original data were discard, due to its simple design. However, part of the conclusions in the current version was totally the same with the previous version (Comparison between SSV10 and STVP). In order to save the animal numbers, only 9 animals were used in the current study, and one animal died during the experimental procedure. These information has been added in the revised manuscript (L97-102).
3. I appreciate, Authors effort to explain sampling order using the flow chart. However, it is still not very easy to understand what the exact experimental setup was. I understand that, sample order is on the timeline, and collection method is shown with the figures. But, at first view all of this is very confusing. E.g. which sample was collected using “Rat holder-awake”? Or what is the difference between Sample I and Sample III collected on day 0 and day 15 (is it just the time between the collections?). Perhaps, it would be more useful to align, figures vertically under the day of collection, so it would be straightforward what procedures were done on which day. Moreover, the manuscript lack consistency with regard to naming the sample groups. For example, in PCA plots (Figure 5) same names are used as in the flow chart (e.g. Sample III). However, in Figures 2,3,4,6 and 7 different naming patter was applied (e.g. SSv0 and STvs). Authors should decide on one naming pattern and apply it consistently through the text.
Answer: Thank you for your careful review. Fig. 1 has been revised according to your suggestions. Furthermore, the naming of the samples was corrected for its consistently through the whole manuscript (L103-108).
4. Plasma is the liquid, cell free, part of blood. Therefore, in order to obtain blood plasma, the use of anticoagulants, such as EDTA, is required. When blood is allowed to clot than and centrifuged, the remaining fluid is serum. Serum and plasma have similar but not identical metabolic composition. Please clarify, if there were anticoagulants used in this study, and if not change the wording, from “plasma” to “serum”.
Answer: Thank you for your professional interpretation. It has been corrected into serum in the revised manuscript.
5. There appear to be some erroneous assignments. For example: Peak 10 is assigned to citrulline, but it looks more like a lipid signal (break resonance). It is more likely the CH2 group of VLDL lipid. Example assignment of could be find in these papers:
a. https://www.ncbi.nlm.nih.gov/pubmed/25213261
b. https://www.ncbi.nlm.nih.gov/pubmed/7762816
Also, how is 3-hydroxyisobutyrate (peak 45) so high at 3.5 ppm, while the doublet at 1 ppm is almost absent? I recommend, verifying the metabolite assignments.
Answer: According to your valuable questions, we have verified the metabolite assignments carefully according to several publications[1-7].
References
1. Feng, J. H.; Li, X. J.; Pei, F. K.; Chen, X.; Li, S. L.; Nie, Y. X., H-1 NMR analysis for metabolites in serum and urine from rats administrated chronically with La(NO3)(3). Analytical Biochemistry 2002, 301, (1), 1-7.
2. Musharraf, S. G.; Siddiqui, A. J.; Shamsi, T.; Choudhary, M. I.; Rahman, A. U., Serum metabonomics of acute leukemia using nuclear magnetic resonance spectroscopy. Sci Rep-Uk 2016, 6.
3. Mika, A. K., Critical evaluation of 1H NMR metabonomics of serum as a methodology for disease risk assessment and diagnostics. Clinical Chemistry & Laboratory Medicine 2008, 46, (1), 27-42.
4. Sheedy, J. R., Metabolite analysis of biological fluids and tissues by proton nuclear magnetic resonance spectroscopy. Methods Mol Biol 2013, 1055, 81-97.
5. Nicholson, J. K.; Foxall, P. J.; Spraul, M., .; Farrant, R. D.; Lindon, J. C., 750 MHz 1H and 1H-13C NMR spectroscopy of human blood plasma. Analytical Chemistry 1995, 67, (5), 793-811.
6. An, Y.; Xu, W.; Li, H.; Lei, H.; Zhang, L.; Hao, F.; Duan, Y.; Yan, X.; Zhao, Y.; Wu, J.; Wang, Y.; Tang, H., High-fat diet induces dynamic metabolic alterations in multiple biological matrices of rats. J. Proteome Res. 2013, 12, (8), 3755-68.
7. Deja, S.; Porebska, I.; Kowal, A.; Zabek, A.; Barg, W.; Pawelczyk, K.; Stanimirova, I.; Daszykowski, M.; Korzeniewska, A.; Jankowska, R.; Mlynarz, P., Metabolomics provide new insights on lung cancer staging and discrimination from chronic obstructive pulmonary disease. J. Pharm. Biomed. Anal. 2014, 100, 369-380.
6. Figure 2B. It is supposed to be averaged NMR spectrum with SEM value. Where are SEM values? Also, in the text (Page 11, line 378-384) there is no reference to the figure. Authors claim that the variability of metabolites is different in different sampling methods, but there is no data to back it up.
Answer: Thank you for your careful review. The color of SEM values of the NMR spectrum were gray in the original manuscript. In order to illustrate it clearly, the color of SEM has been changed into red. Furthermore, the reference to the figure has been added in the revised manuscript (L219). We also added the discussion of the variability of the metabolites in the revised manuscript (L219-224).
7. Figure 4. This figure is very useful, but it could be improved. In particular, the regression line calculated from the data (the one that result in R2 values presented in the plots) should be plotted in addition to the y=x line.
Answer: That is a good suggestion. The regression line has been added in the revised manuscript.
8. Figure 5, the number of samples in groups is very suspiciously. It is stated in the text, that there were 8 animals in total (Page 2, line 137) and samples were collected from these 8 animals at different times, using different methods. However, upon investigation of Figure 5 these are the numbers of samples:
a. Fig5A: Sample I n= 9, Sample III n=8
b. Fig5B: Sample III n=7, Sample IV n=6
c. Fig5C: Sample V n=6, Sample VI n= 8
d. Fig5D: Sample I n=9, Sample II n=9
e. Fig5E: Sample V n=7, Sample III n =8,
f. Fig5F: Sample II n=9, Sample V n=7
My calculations for some of n<8 numbers may not be accurate due to points overlap and actually all 8 samples were used for plotting. However, there is no excuse for having more points than are supposed to be in the group. Please clarify, what happened here.
Answer: Thank you so much for your careful review. Yes, the animal number was nine in total. However, one animal was dead during anesthesia procedure. Furthermore, the blood collection operation was ceased after three attempts on an animal. Thus the sample number was varied from 6 to 9, which has been added in the revised manuscript (L97-102).
9. Page 1, line 40, cell culture media is not a body fluid.
Answer: Yes, it has been deleted in the revised manuscript.
10. Page 4, line 244, change: “The mixtures” to “The samples”.
Answer: It has been corrected into ‘The samples’.
11. Page 5, line 316, change: “A sample” to “An example”.
Answer: It has been changed into ‘An example’.
12. Please explain how the concentrations for quantified metabolites (Figure 7) were obtained.
Answer: It has been explained in the revised manuscript (L163-168).
Round 2
Reviewer 2 Report
The Authors addressed all of my major comments. I have just few minor issues with the current version of the manuscript:
- Page 6, Table with assignments should be numbered “Table 2”
- Page 12, line 302-303, the Table with R2X and Q2 parameters of chemometric models is missing.
- Page 12, line 318, citrulline is still mentioned in the text, although, after last round of revisions it has been removed from the list of assigned metabolites.
- Figure 7, in the OPLS-DA score plots, axis is labeled the same as in the PCA. Please check, but it may need to be changed from PC1 and PC2 to t1 score and to1 score.
- Page 14, lines 346-348, the sentence “When the concentration of insulin decreases, the hepatic glucose homeostasis shifts from glycogenolysis/gluconeogenesis to glycogen synthesis via insulin signaling, which increases the concentration of glucose.“ is incorrect. Liver releases glycogen and upregulates gluconeogenesis under low insulin high glucagon conditions. Glycogen storage is activated upon feeding when inulin action is increased. Additionally, it is not clear if this sentence describes concentration of glucose in the liver or in the blood. Please correct this.
Author Response
Question 1: Page 6, Table with assignments should be numbered “Table 2”
Answer: It has been corrected into ‘Table 2’ (L214).
Question 2: Page 12, line 302-303, the Table with R2X and Q2 parameters of chemometric models is missing.
Answer: Table 3 has been added in the revised manuscript (L307).
Question 3: Page 12, line 318, citrulline is still mentioned in the text, although, after last round of revisions it has been removed from the list of assigned metabolites.
Answer: Thank you for your careful review. It has been deleted in the revised manuscript (L321-323).
Question 4: Figure 7, in the OPLS-DA score plots, axis is labeled the same as in the PCA. Please check, but it may need to be changed from PC1 and PC2 to t1 score and to1 score.
Answer: It has been corrected into t[1]P score and t[2]O score (L331-332).
Question 5: Page 14, lines 346-348, the sentence “When the concentration of insulin decreases, the hepatic glucose homeostasis shifts from glycogenolysis/gluconeogenesis to glycogen synthesis via insulin signaling, which increases the concentration of glucose.“ is incorrect. Liver releases glycogen and upregulates gluconeogenesis under low insulin high glucagon conditions. Glycogen storage is activated upon feeding when inulin action is increased. Additionally, it is not clear if this sentence describes concentration of glucose in the liver or in the blood. Please correct this.
Answer: Thank you for your careful review. This sentence has been corrected into ‘When the concentration of insulin decreases, the hepatic glucose homeostasis shifts from glycogenolysis/gluconeogenesis to glycogen synthesis via insulin signaling, which increases the concentration of glucose in blood [29].’ (L348-350).
This manuscript is a resubmission of an earlier submission. The following is a list of the peer review reports and author responses from that submission.
Round 1
Reviewer 1 Report
the authors applied NMR-based profiling to differentiate the rat plasma from two different sampling methods: blood from saphenous vein in the awaken state, and from inferior thoracic vena cava in the anaesthetized state. Overall both the scientific conclusion and write-up are weak. A significant revision should be considered before publishing.
-A prime concern on the concluding results (and must be addressed): the metabolic differences found in the study could also attribute to the fact that the two-comparison blood groups are sampled from the different part of the rat (saphenous vein vs inferior thoracic vena cava). Therefore, the authors must be caution on interpreting the data (i.e. OPLS data).
One should consider normalizing the factors that contributed to the different body regions. (i.e. collect samples from the SAME body regions).
-Blood collection, line 95-101: no explanation was given? How are these observations affect the results in the study, especially the blood metabolome? Why ipsilateral and contralateral?
-line 131: PLS-DA is mentioned, but there are no data or results are discussed in the manuscript?
-NMR-assignments: provide at least one NMR reference
-Figure 1 and its entire text are unclear! First, the use of two different sampling groups from saphenous veins is unclear? Clarification is needed.
-line 186: what is OSC-filtered NMR data?
- lines 188-190 not relevant in your data since the score plot shown no outliners.
-Fig 2, where is the loading plot that stated in line 196?
-Fig 2. why separate the sample-groups of saphenous vein? It is more appropriate to compare all data from saphenous vein and thoracic vena cava.
-The OPLS results might not reflect what the authors’ conclusions (i.e. 1st comment)
English need to be improved
-use formal text
-eliminate long sentence structures (e.g. lines 163-176)
-line 109, remove ‘serum’… the study is on plasma samples not serum.
Reviewer 2 Report
The manuscript by Du and Li et al. investigates perturbations in metabolic profiles of blood samples collected from rats with and without anesthesia. The Authors utilize 1H NMR spectroscopy and chemometric analysis to compare paired datasets. While, the design of this study is acceptable, its main limitation is a lack of scientific novelty. The metabolic effects of anesthesia have been known for many years and studied in context of blood and tissue sampling. It is widely accepted, that the use of anesthesia will result in significant perturbations in the metabolic composition of blood due to alterations in hormonal regulation, decreased blood flow and lowered breathing rate. All these changes affect tissue metabolism. Therefore, the main conclusion of this paper, that “anesthesia could significantly increase glucose concentration and decrease lactate concentration” is not surprising and very much expected. Probably, it would be more interesting if the Authors focused on metabolites other than glucose and lactate.
Major comments:
1. The fact that anesthesia affects glucose blood levels is well established:
https://www.ncbi.nlm.nih.gov/pubmed/16332272
https://www.ncbi.nlm.nih.gov/pmc/articles/PMC4908499/
https://www.ncbi.nlm.nih.gov/pmc/articles/PMC4160585/
Anesthesia effects were also studied in context of tissue sampling:
https://www.ncbi.nlm.nih.gov/pmc/articles/PMC3130322/
https://journals.plos.org/plosone/article?id=10.1371/journal.pone.0117232
and, standard operating procedures when handling rodents:
https://www.ncbi.nlm.nih.gov/pmc/articles/PMC2938392/
2. Total area normalization was used prior to chemometric analysis. This normalization technique can produce biased results as it is affected strongly by highly abundant metabolites (and those having multiple resonances). It is recommended that the Authors use probabilistic quotient normalization (PQN) rather than total are normalization. PQN is advantageous particularly, when large changes in blood glucose concentration occur (see Figure 5 in https://www.ncbi.nlm.nih.gov/pubmed/16808434). Changing the normalization method to PQN should allow detection of more subtle differences in metabolic profiles and reduce an impact of glucose fluctuation.
3. The Authors discuss changes in metabolite levels only in the context of chemometric analysis. However, OPLS-DA is just a tool for selecting candidate metabolites. These metabolites should be further quantified, and statistically tested in order to draw conclusions.
4. Only a single anesthesia method was tested (sevoflurane). Investigation of different commonly used anesthetics on the blood metabolic profile could increase the impact of this study.
5. CPMG details are missing. What was spin-echo delay, number of loops and relaxation delay?
6. In the PCA score plot, PC1 explains more than 90% of variance, while PC2 explains only 2-6% of variance. Thus, almost all information was captured by PC1. Could this be due to the scaling approach used in SIMCA? Authors used “Ctr standardization”, however, this is an uncommon way of analyzing binned data NMR. Rather standardization (Centering of the data) together with Pareto Scaling or Unit Variance (UV) scaling is typically used.
7. Manuscript requires major English proofreading.
Minor comments:
1. Line 108. It should be 1H NMR not H NMR.
Table 1 – It is hard to believe that deoxycytidine could be detected in blood with use of 1H NMR spectroscopy. According to HMDB concentration of deoxycytidine in human blood is around 0.2 μM (http://www.hmdb.ca/metabolites/HMDB0000014) which would be very hard to detect. Could this be some other more abundant metabolite?